# Accurate Adapter Information Is Crucial for Reproducibility and Reusability in Small RNA Seq Studies

**DOI:** 10.3390/ncrna5040049

**Published:** 2019-10-28

**Authors:** Xiangfu Zhong, Fatima Heinicke, Benedicte A. Lie, Simon Rayner

**Affiliations:** 1Department of Medical Genetics, Oslo University Hospital and University of Oslo, 0450 Oslo, Norway; fatima.heinicke@medisin.uio.no (F.H.); b.a.lie@medisin.uio.no (B.A.L.); 2Hybrid Technology Hub—Centre of Excellence, Institute of Basic Medical Sciences, University of Oslo, 0372 Oslo, Norway

**Keywords:** adapter trimming, adapter, small RNA, microRNA, NGS, reproducibility, reusability

## Abstract

A necessary pre-processing data analysis step is the removal of adapter sequences from the raw reads. While most adapter trimming tools require adapter sequence as an essential input, adapter information is often incomplete or missing. This can impact quantification of features, reproducibility of the study and might even lead to erroneous conclusions. Here, we provide examples to highlight the importance of specifying the adapter sequence by demonstrating the effect of using similar but different adapter sequences and identify additional potential sources of errors in the adapter trimming step. Finally, we propose solutions by which users can ensure their small RNA-seq data is fully annotated with adapter information.

Small non-coding RNAs comprise short RNAs less than 200 nt in length, including microRNAs (miRNAs), Piwi-interacting RNAs (piRNAs) and small nucleolar RNAs (snoRNAs) and which have a variety of functions. Next generation sequencing (NGS) is the most common technique for studying small RNA expression. An NGS experiment requires the construction of a library, which includes ligation of an adapter that acts as a binding site for priming the sequencing reaction and capturing the endogenous small RNA inserts [1,2]. Due to the short insert size, the 3’ adapter sequence is commonly included in the raw data reads and removing the adapter from raw reads is a necessary data analysis step. Most studies consider adapter trimming a straightforward pre-processing step. For other NGS experiments such as RNA-seq and ChIP-seq, the insert size is generally longer and the adapter trimming step has less impact. However, there are many issues that can affect the trimming output, which might influence further downstream analysis. For example, under- or overtrimming a read can make a significant difference in the quantification of small RNAs of interest. The problem is further exacerbated when mismatches are allowed during mapping. However, an even more serious challenge is ensuring that the adapter sequence is available, or correctly specified. For example, the adapter sequence in the instruction manual for the NEBNext Small RNA Library Prep Kit for Illumina is specified as:
5’-rAppAGATCGGAAGAGCACACGTCT-NH2-3’
where the *rApp*, a 5’-adenylated termini, is removed when the 3’ adapter and inserted RNA fragment are ligated by the T4 RNA ligase [3].

However, in several studies, for example, [4,5,6] the *rApp* was replaced with an additional Adenine so the 3’ adapter sequence was specified as:
**A**AGATCGGAAGAGCACACGTCT

It is unclear whether this modified adapter sequence was used for adapter trimming, but it highlights how mistakes can occur and be propagated in subsequent publications.

Many tools are available for 3’ adapter trimming such as cutadapt [7], fastx_clipper (http://hannonlab.cshl.edu/fastx_toolkit/) and Trimmomatic [8] and they generally require a specified adapter sequence as mandatory input. However, while data portals such as The Gene Expression Omnibus (GEO) and Sequence Read Archive (SRA) require deposition of original raw data files, they do not require adapter information as mandatory data. Thus, several tools have been developed to identify adapter sequence from the raw sequencing data, e.g., minion (part of the kraken package [9]), DNApi [10] and AdapterRemoval v2 [11]. However, these predicted adapter sequences should be used with care.

Thus, there are two situations that may lead to incorrect specification of adapter sequence in read trimming. Firstly, different vendors use distinct or similar 3’ adapter sequences in their small RNA sequencing library preparation kits (Appendix A). Moreover, some kits have been discontinued or outdated. For example, the ScriptMiner Small RNA-Seq Library Preparation Kit from Epicentre was discontinued in December 2013 and version 1 of the TailorMix kit is outdated. Secondly, adapter information is not always available. As mentioned above, the GEO and the SRA do not require adapter related data as mandatory information. A report in 2016 found that only one third of NGS entries in the GEO provide adapter information [10]. Moreover, even when the library preparation kit is specified, it is not always straightforward to find the corresponding adapter sequence. For example, NEBNext, NEXTflex and TruSeq specify the sequence in their product instruction manuals; Lexogen and QIAseq provide the information on their websites; the CleanTag 3’ adapter sequence is specified in a publication [12]; and the TailorMix, the SMARTer and the TrueQuant (used by GenXPro) adapters are declared neither on their websites nor manuals. Additionally, incorrect kit information is sometimes given in publications. For example, some studies have reported preparing libraries using the TailorMix miRNA Sample Preparation Kit v7 [13,14] instead of the currently available kit v2 (as of August 2019, https://www.seqmatic.com/products/tailormix-mirna-sample-preparation-kit-v2/).

To review the situation in 2019, we retrieved small RNA-seq data entries from the SRA by searching for “miRNA-Seq[Strategy]” and obtained 41,607 entries (access date 27 May 2019). TruSeq is the most used kit (although 335 experiments specified this as *TrueSeq*) followed by NEBNext (see Appendix A). However, still less than half (47%) of entries provide kit information. For the 53% SRA entries are lacking this data, software tools are needed to determine the best guess for the adapter sequence to proceed with trimming. As a simple test, we searched for the adapter in the data from [15] using *minion* [9], *DNApi* [10] and *AdapterRemoval* [11]. Of these tools, only *DNApi* gave the correct, but partial, sequence. The results are shown in Appendix A.

To determine the impact of adapter sequence and trimming protocol on trimming results, we used data from a study by Dard-Dascot et al. [15]. For details, see “Materials and Methods” in the Appendix A. The goal of this study was to identify bias in various library kits by resequencing the same samples using different kits. One sample included six synthetic RNAs, RNA1 to RNA6, of known sequence (see sequences in Appendix A, from [15]) and these were the focus of our study. Firstly, we investigated the effect of using three similar adapter sequences on the dataset prepared by the NEBNext kit. The three sequences were the correct adapter sequence from the kit manual (NEBNext_trim01), the sequence with an additional A at the 5${}^{\prime }$ end (NEBNext_trim02) and the partial adapter sequence from the CATS trimming instructions (NEBNext_trim03) with two nucleotide changes relative to the correct adapter sequence (Figure 1A). The results are shown in Figure 1B–D. Figure 1B shows the effect of using the three different adapters (NEBNext_trim01, NEBNext_trim02 and NEBNext_trim03) on the read count of RNA1 to RNA6 in terms of the percentage of perfectly matched reads identified in the trimmed data. RNA6 is not detected by any of the kits, suggesting an issue with sequencing rather than a trimming problem. When the correct adapter, NEBNext_trim01, is used all the RNAs are detected to varying degrees. RNA1, RNA2 and RNA5, are the three most highly represented—see Appendix A for counts—with more than 90% of these RNAs perfectly trimmed from the raw data. However, when the highly similar NEBNext_trim02 and NEBNext_trim03 adapters were used, fewer than 1% of the reads were correctly trimmed. In this case, an analysis would fail to detect the presence of these RNAs.

To further confirm that this is not a kit-specific effect, we also tested the TruSeq dataset with eight different adapter sequences including the correct TruSeq 3’ adapter sequence. We found the same results as the NEBNext dataset. Any additional or removed nucleotide at the 5’ end of the adapter gives a different trimming result (Appendix A). Thus, the correct adapter sequence is required for correct read trimming, even a single base difference in the adapter sequence can introduce large changes in the digital gene expression (Figure S1B). Similar results are also found within the data from [6], see Appendix A.

However, providing the correct kit information does not always guarantee correct adapter trimming because of specific procedures in the library preparation protocol. For example, in the CATS Small RNA-seq protocol, single stranded RNAs are first polyadenylated at the 3’ end, and then cDNA synthesis is performed in the presence of a poly(T) anchored adapter. Therefore, in addition to the adapter sequence, the poly(A) between the RNA insert and adapter needs to be properly removed. Figure 1C shows the result of trimming the raw data prepared with the CATS Small RNA-seq Kit by simply specifying the provided adapter sequence. In this case, virtually no reads were retrieved after trimming. Once again, failure to following the manufacturer’s protocol for adapter trimming will impact the trimmed output. Similar problems will arise in the dataset prepared by NEXTflex, which requires removal of four additional random nucleotides after adapter removal. Few adapter trimming tools provides built-in functions to handle these additional nucleotide removal steps.

Figure 1D shows the results from following the detailed trimming instructions provided in the CATS manual using three different versions of the user manual dated January 2017, March 2017 and September 2017. The instructions in the first two versions are identical, and detect notably more reads than following the instructions in the latest version of the manual, only read RNA5 is consistently reported after all three procedures. Thus, reporting the kit version is necessary to allow users to reproduce the trimming, however this information is not always provided, for example, in [15].

Additionally, the provided trimming instructions are not effective for the two synthetic RNAs that have an adenine at the 3’ end (RNA2 and RNA4 in this case). The perfectly trimmed reads from these two RNAs were relatively few with less than 10 untransformed raw counts, despite thousands of untrimmed reads present in the raw data (see Appendix A).

Thus, analysis of sequencing data is only reproducible with access to accurate and unambiguous specification of the adapter sequence that was used in library preparation. Missing adapter sequence represents a major obstruction to taking advantage of the vast amount of publicly available data. Providing comprehensive and consistent information regarding the adapter sequence benefits the research community in general. We therefore propose the three following solutions to increase the reproducibility and accuracy in small RNA-seq studies.

Firstly, most journals publishing articles on sequencing analysis require that users deposit accession numbers to public databases such as the SRA, GEO or ArrayExpress. At the same time, editors could require the adapter sequence to be specified in submitted small RNA sequencing studies. Researchers can also take responsibility by ensuring they include all relevant detail about the adapter, as well as kit information.

Secondly, the fastest and most straightforward approach is to include adapter information as mandatory metadata which needs to be filled out as part of the process of submitting sequencing data to public data repositories such as the SRA. Implementing the FAIR (findable, accessible, interoperable, reusable) principles in data repositories could maximize the sharing and reproducibility of sequencing data [16]. The detailed information regarding library kit version and adapter sequence should be included at this point.

Thirdly, consolidated information from kit manufacturers, offering practical adapter trimming advice, as well as archived versions of adapter sequences, would also be helpful to the research community. Biases, including barcoding, structural and ligation bias are well known and have been discussed in the context of small RNA sequencing [17,18,19,20,21]. Manufacturers are continuously developing or revising kits to achieve more efficient ligation, reduce bias and improve performance, e.g., the NEBNext and NEXTflex kits are currently at version 3, and the current TailorMix kit is version 2. Any information about adapter updates is essential for users and information should be readily available. As poly(A) based approaches are becoming more popular in library preparation, protocols for poly(A) removal should also be defined for accurate adapter trimming. Many small RNAs (such as miRNAs) contain adenines at their 3’ end and these may be mis-trimmed as part of poly(A) and lead to mis-quantitation. The CATS kit from Diagenode provides ready-to-use trimming instructions in their manuals, and these were updated when they renewed their kit. Nevertheless, our simple analysis indicates that trimming these sequences ending in one or more adenines remains a challenge.

Based on their crucial role in small RNA-seq data analysis, adapter information should follow findable, accessible, interoperable and re-usable (FAIR) principles [22]. The reality is that accurate adapter trimming is not a straightforward process. Consequently, more attention needs to be paid to this step to ensure the correct sequence and protocol is used. Access to correct and detailed adapter information will help to minimize some of the problems associated with this issue.

## Figures and Tables

**Figure 1 ncrna-05-00049-f001:**
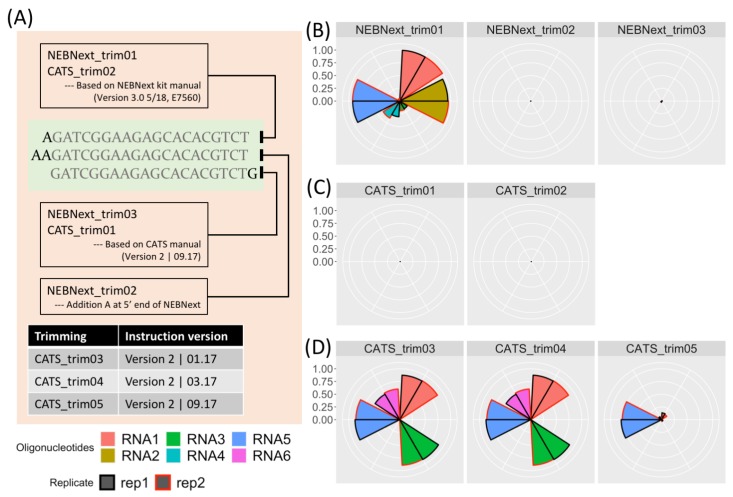
Use of incorrect adapter sequence or trimming protocol can lead to incorrectly trimmed reads and miscounting of reads mapping to features. The selected datasets are originally from [15], samples containing six synthetic small RNAs prepared by the NEBNext and CATS kits. After trimming, the Linux command *grep -c* was used for counting. (**A**) Top part of figure: Schematic of the different but highly similar adapter sequences used in (**B**–**D**). Bottom section of figure: Different versions of the CATS manual used for trimming protocol applied in (**D**). Legend under (**A**) Fill colour corresponds to the six synthetic RNAs in the dataset, line colour corresponds to the two replicates for each sample. Sequence for oligonucleotides are listed in Appendix A. (**B**) Choice of adapter sequence can have a major impact on downstream analysis. Left: Use of the correct adapter sequence (NEBNext_trim01) identifies the presence of 5 out of 6 synthetic small RNAs present in the NGS dataset. Middle and right (NEBNext_trim02 and NEBNext_trim03): Using a highly similar adapter sequence that differs by one or two nucleotides has a drastic effect on mapped reads with less than 1% of reads identified. (**C**) In some case detailed trimming instructions are required in addition to the adapter sequence. The trimming sets CATS_trim01 and CATS_trim02 were trimmed by specifying the correct adapter sequence, but few perfectly trimmed reads were detected. (**D**) The problem extends to incorrect application of manufacturer’s protocol during read trimming. From left to right, trimming results after following trimming instructions specified in the January 2017, March 2017 and September 2017 releases of the manual. The instructions in the latest version are distinct from those provided in the previous two versions and this is reflected in the number of identified reads, with the latest protocol identifying notably fewer reads associated with the synthetic RNAs. CATS_trim01 was trimmed using the same adapter sequence as CATS_trim05, demonstrating that for some kits, specifying the adapter alone is not sufficient to achieve efficient read trimming.

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
