# Peer review of "Accurate Adapter Information Is Crucial for Reproducibility and Reusability in Small RNA Seq Studies"

_ncrna, 2019, doi:10.3390/ncrna5040049_

Round 1

Reviewer 1 Report

The commentary “Accurate adapter information is crucial for reproducibility and reusability in small RNA seq studies” is an interesting overview of the potential errors that can result from incorrect adapter sequence pre-processing during small RNA-seq data analysis. Some minor comments:

-There is a disconnect between the main text (commentary) and the supplementary materials that makes it hard to understand them both as a unit.  For example, Table S5, S6, and S7 are not referenced in the commentary. I would suggest incorporating lines 2-19 from supplementary materials into the main text, as well as removing the presence of duplicated references (Dard-Dascot et al. and Gümürdü, A. et al.). A rearrangement of supplementary tables and figures would also be required. 

-The first line in the abstract is redundant.

-Line 24: I would suggest adding “for Illumina” after the name of the kit.

-Line 31: remove “in.”

-Line 39: For clarity, I would suggest rewriting the sentence “However, as these return best guesses they should be used with care.”

-Line 68: You should reference the sequences of the six synthetic RNAs (Supplementary Table S7) here, where you first introduce them. 

-Line 124: “maximise” check spelling.

-In Table S6, I would recommend moving the first column (reference) to the end, as well as adding an extra column with the name of the first author.

-I would also recommend adding more information at the top of each table header to describe in more detail the rows below.

-In general, there is a lack of consistency throughout the text (example poly(A)/Poly(A)/polyA) that needs to be addressed.

Reviewer 2 Report

Small RNA highthrougput sequencing has gained momentum during the last decade, profiting of the continuous advances in the field of 2nd and 3rd generation sequencing as well as the improvements in molecular biology. As a consequence of the required heavy library preparation steps, this data requires rigorous preprocessing steps to ensure reproducibility and accuracy in the biological conclusion drawn. Among these preliminary steps, one of the most crucial consist in trimming the sequence of adapters that have been ligated to the matrices of interest during the library construction. Despite being determinant, this step is still often overlooked by researchers and poorly documented in studies relying on this technology.

In this technical report, Zhong et al. illustrate and quantify the impact of wrongly / poorly defined adapter sequence during the adapter trimming step on the quality of small non-coding RNA sequencing data. Using several publicly available datasets generated from libraries (synthetic small RNA) built with different kits (NEBNext, TruSeq), they demonstrate how as little as a single extrabase in the adapter sequence provided to the trimming algorithm lead to spurious conclusions (miscounting or no detection at all). They results further emphasize the urgent need for transparency when reporting results drawn from sequencing data (with an emphasize on small RNA-seq) and for a broad commitment of the scientific community to adopt FAIR principles.

The manuscript is really well written, informative, comprehensive and pleasant to read. I do think that the scientific community will really benefit of such an intelligible and obvious technical study to further avoid “easy-to-deal-with” artifacts and improve reproducibility and transparency in biological science.

Even if slightly out of scope, I do think that briefly discussing about the impact of wrongly / poorly designed adapter trimming step on other NGS-derived data (RNA-seq, ChIP-seq …) would be an interesting way to wrap-up the story.

In addition, throughout the manuscript, the authors rely on a simplistic counting approach (with the UNIX command grep -c) for matching trimmed sequences to the known miRNAs. What is the impact of a poorly designed trimming strategy with wrongly / poorly defined adapter sequence in a realistic workflow (i.e. including an alignment step on a reference genome or miRNome / transcriptome) ? How the choice of the aligner and parameters (end-to-end or seeded alignment) would balance / magnify this impact ?
